Journal of Machine Learning Research 23 (2022) 1-11        Submitted 1/21; Revised 5/22; Published 9/22

# Upscaling Prostate Cancer MRI Images to Cell-level Resolution Using Self-supervised Learning

**Yaying Shi**                                                                    yshi10@uncc.edu
**Srijan Das**                                                                   sdas24@charlotte.edu
**Yonghong Yan**                                                                 yyan7@charlotte.edu
*College of Computing and Informatics*
*University of North Carolina at Charlotte*
*Charlotte, NC 28262, USA*

**Editor:** My editor

## Abstract

Magnetic Resonance Imaging (MRI) plays a pivotal role in medical imaging, particularly in the diagnosis and treatment of cancers via radiography. However, one of the limitations of MRI is its low spatial resolution, which can hinder the accurate detection and characterization of cancerous lesions, especially those that are small or subtle in nature. There is a growing need for advancements in MRI technology to improve the resolution of MRI, particularly in the field of oncology, where precise detection and segmentation of tumors are crucial for effective treatment planning and optimal patient outcomes. In this paper, we proposed a self-supervised deep learning technique to upscale cancer MRI images to cell-level resolution with pathology Whole Slide Imaging (WSI). By integrating information from pathology WSIs with MRI images, this approach aims to create hybrid images that offer a more detailed and comprehensive view of cancer tissue structures. We evaluated our techniques using prostate lesions both on the similarity metrics and downstream segmentation tasks. For the similarity, our reconstructed fusion images can achieve an average 0.933 in structural similarity index. We improved lesion segmentation dice score from 57.3% to 64.0% on the test cases. Such fusion of the two imaging modalities shows promise for improving the accuracy and reliability of cancer diagnosis, guiding treatment decisions, and ultimately improving patient outcomes.

**Keywords:**  MRI, Pathology, Self-supervised Learning, Image Registration.

## 1 Introduction and Background

Radiological imaging constitutes a cornerstone in the study of cancer, spanning critical stages from foundational research to diagnostic elucidation, therapeutic strategizing, and ongoing surveillance. Modalities such as computed tomography (CT), magnetic resonance imaging (MRI), and positron emission tomography (PET) furnish intricate depictions of internal anatomical structures, affording clinicians invaluable insights into tumor localization, metastatic dissemination, and anomalous tissue proliferation.

The interpretation of radiographic imagery presents challenges, particularly in distinguishing between malignant and benign tissue, which can be subjective even for experienced experts. Manual demarcation of cancerous lesions on radiological scans, though essential, often introduces inaccuracies that may underestimate tumor dimensions or miss less con-

spicuous lesions due to low image resolution. In contrast, pathology whole slide images (WSI) offer gigapixel-level resolution, providing pathologists unprecedented insight into cellular morphology, tissue architecture, and aberrant features with exceptional precision. By aligning histopathology images with corresponding radiological slices, clinicians can overlay cancerous regions identified from histopathological analyses onto radiological scans. This approach enables precise tumor segmentation, including lesions not easily visible on MRI scans, thereby enhancing cancer evaluation.

However, fusing images from different modalities presents technical hurdles due to inherent disparities in resolution, particularly when integrating images of large resolution gaps such as pathology and radiology images. Currently, image registration techniques are primarily developed for medical images with similar resolutions, such as PET, CT, and MRI scans Hering et al. (2022); Wang et al. (2020); Sokooti et al. (2017). These methods typically fall into two categories: traditional approaches, which often suffer from low computational efficiency, and machine learning-based methods. The machine learning-based methods are limited in their applicability as they tend to work only on specific datasets and struggle to extend to larger resolution gaps, such as between MRI and pathology WSI. Furthermore, these machine learning approaches face challenges due to the lack of paired and well-labeled data available for training purposes.

To address the limitations of existing machine learning-based registration methods and enhance the generation of fusion images with improved diagnostic capabilities, we present a novel self-supervised learning framework specifically designed for the registration of radiological and pathological images. This is achieved through the utilization of a self-supervised transformer-based feature extraction network and a feature-matching network. Our approach is able to enhance the resolution of MRI images to the cell level using pathology data, with the registration process serving as a crucial tool in achieving this goal. By leveraging advanced self-supervised learning, our framework overcomes the constraints of current registration methods which need multiple paired and well-labeled data. Our contributions be summarized as follows:

1. We developed a novel image fusing network for upscaling prostate MRI images to cell-level resolution, leveraging registration as a key tool. The framework enables the creation of fused images that seamlessly integrate high-resolution pathology data with MRI scans. We have achieved 39 times resolution enhancement between the original MRI and the new fusion image. Using the fused images for downstream tasks such as cancer segmentation, it can offer enhanced details and accuracy crucial for improving diagnostic capabilities.

2. We evaluated our framework using prostate cancer datasets, comparing it with the original paper that presented those datasets Shao et al. (2021). Our framework demonstrated an enhancement in accuracy from 56.3% to 64.6% for prostate cancer. We also evaluated the similarity of our new fused image and up-scaled MRI image by Structural Similarity Index (SSIM). We achieved an average SSIM of 0.939 and a minimum of 0.933, suggesting that our new fused images have high similarity to the MRI images.

3. We tackled the issue of insufficient labeled or paired pathology and radiology images using self-supervised learning, which enables model learning using unlabeled datasets. The efficiency of the pre-trained self-supervised learning can be seen in the Appendix C.

## 2 Proposed Framework

For medical images, which are multimodal in nature, fusion and registration of images of different modality have been well studied topics. Our method advanced the state-of-the-art by using self-supervised learning, which alleviates the dependency on labeled data by leveraging the inherent structure and redundancy within the data itself. The self-supervised learning framework is inspired by DINO Caron et al. (2021) and Prosregnet Shao et al. (2021). In contrast to DINO/Prosregnet that are mainly used for image classification, ours is an image fusion network for images that have high-resolution gaps, such as the tissue-level MRI and the cellular-level pathology images, as illustrated in Fig. 1. The key component of the network includes 1) Two self-supervised feature extractors, each devoted to MRI and whole-slide pathology images respectively. 2) A correlation mapping block responsible for generating correlation maps of features extracted by the previous feature extractors. 3) A feature-matching sub-network designed to align and map distinctive features extracted from both image types. 4) Post-processing techniques for fusing pathology and MRI patches based on the correlation maps obtained in the previous step.

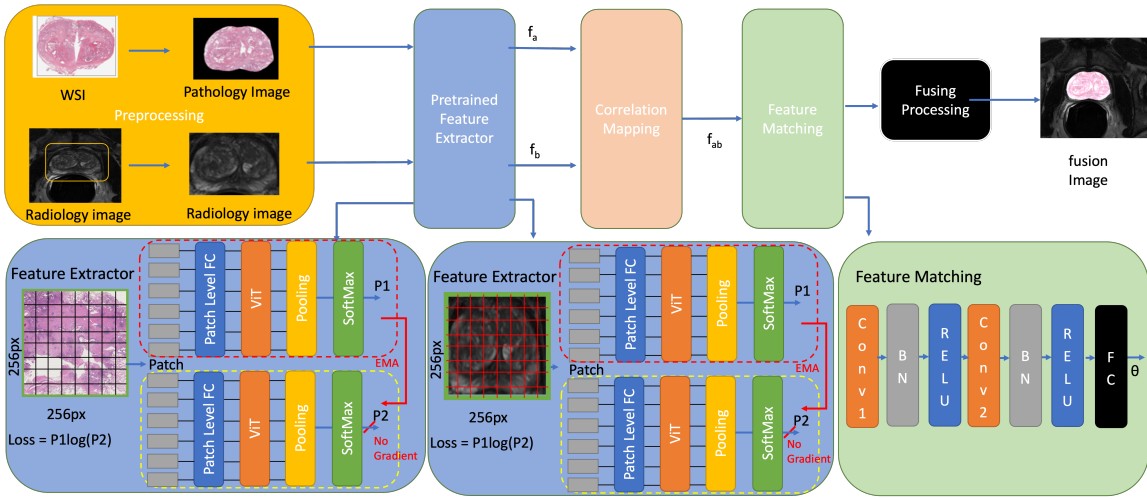

Figure 1: Overall Core Architecture Design of Cell Level Precision Registration

### 2.1 Data and Pre-processing

In this study, MRI and pathology data were obtained from TCIA Clark et al. (2013) and TCGA for Cancer Genomics. TCIA provided two datasets of paired MRI/pathology data, extensively described in the original work Shao et al. (2021). The first dataset, PROSTATE-MRI P (2016), consists of 26 cases, with each case containing multiple pathology slides. In total, we have 82 paired MRI and Pathology WSI from this dataset. The second dataset Madabhushi (2016), referred to as fused MRI-Prostate, comprises 28 cases, each containing 3 Tesla T1-weighted, T2-weighted, Diffusion weighted, and Dynamic Contrast-Enhanced prostate MRI scans, accompanied by corresponding digitized histopathology (H&E stained) images of radical prostatectomy specimens. For training purposes, we utilized all images from the first dataset and 26 cases from the second dataset. The remaining

two cases, along with 6 slides annotated with lesions, were reserved as the test dataset. The first dataset was not used for the test cases due to the absence of lesion annotations necessary for evaluating downstream segmentation tasks. Given the limited size of the datasets, comprising only 82 paired data points, we relied on machine learning-based methods. To mitigate the data limitations, we extended the training data by incorporating additional unlabeled MRI and pathology images from TCIA Archive and TCGA for Cancer Genomics, respectively. Each pre-training dataset consisted of 500 cases, enabling us to pre-train a feature extractor in a self-supervised manner.

In the initial stage, we trained the feature extractor using unlabeled MRI and pathology image data. We subdivided the acquired WSI from TCGA and MRI scans from TCIA into smaller patches measuring 256 * 256 pixels. For the subsequent step, we trained our feature matching subnet utilizing two paired MRI/pathology datasets. We standardized the dimensions of these datasets to match those of our pre-trained feature extraction data. Because the data of the two datasets are paired, corresponding MRI slides and pathology images were aligned, as depicted in the $PathologyImage$ and $RadiologyImage$ sections of Fig. 1. Furthermore, when performing downstream tasks on both pathology slices and MRI scans, such as lesion annotation, we applied the same methodology. This ensured alignment and correspondence between the annotations, as illustrated in the $RadiologyImage$ section of Fig. 1.

## 2.2 Self-supervised Feature Extractor Based on DINO

The first component of our framework comprises a self-supervised feature extractor inspired by DINO Caron et al. (2021). DINO is a self-supervised learning method for visual representation learning. It achieves cutting-edge performance by aligning representations of the same image across different layers of the neural network through self-distillation. In this paper, the initial phase involves pre-training a pathology and MRI feature extractors by leveraging the DINO framework. Given the disparity in resolution and format between MRI and pathology images, we undertake separate pre-training processes for each modality. Consequently, we develop distinct feature extractors tailored to MRI and pathology images, as illustrated by the two feature extractors depicted in Fig. 1.

To optimize performance, we leverage pre-trained weights obtained from the TCIA dataset for both the student and teacher networks, as outlined in Caron et al. (2021). Given the larger size of pathology images compared to natural images, we decompose each image into smaller patches, each measuring $256 \times 256$ pixels. These patch images are fed into both the student and teacher networks, which share identical network structures. The networks employed standard vision transformer blocks, depicted in the feature extractor block at the bottom of Fig. 1, featuring components such as Layer Normalization, multi-headed self-attention, residual connections, and a multi-layer perceptron. We adopted the ViT-small as the backbone for our feature extractor, maintaining the parameter settings as per Caron et al. (2021). During the pre-training phase, the teacher network is frozen with weight updates solely in the student network. A distillation loss across the teacher-student predictions is imposed to train the self-supervised framework (see Fig. 1). The parameters in the teacher network is updated using Exponential Moving Average (EMA).

## 2.3 Feature Matching Sub-Network Based on CNN

Our feature-matching network, inspired by the methodology outlined in the work of Shao et al. Shao et al. (2021), consists of two main components: correlation mapping and feature matching. Initially, we obtained features for both MRI and pathology images from two separate feature extractors. Each feature map, denoted as $f$, represents an image with dimensions $(w, h, d)$, where $d$ represents the number of features, $w$ represents the width, and $h$ represents the height. Subsequently, the feature maps $f_A$ and $f_B$ were downsampled into a smaller dimension representation to reduce computational costs. These downscaled feature maps were then input into a correlation layer, which computes the dot product of input features to quantify the similarity between the two images. This correlation layer combines $f_A$ and $f_B$ to generate a correlation map $C_{ab}$ of the same size. The computation of the correlation map is expressed as:

$$C_{ab}(i, j, k) = f_B(i, j)^T * f_A(i_k, j_k) \tag{1}$$

The equation $k = h(j_k - 1) + i_k$ is used to calculate the index variable $k$ based on the indices $i_k$ and $j_k$, with $h$ representing the width of the feature map. The resulting correlation map $C_{ab}$ indicates the similarity of features from $f_b$ at position $(i, j)$ with all features from $f_a$. To address potential ambiguous matches, normalization is applied to obtain the correspondence map $f_{ab}$. This map then undergoes processing by a feature-matching network, which is a regression network responsible for estimating the parameters of the geometric transformation associated with the input MRI and pathology images. Following the architecture described in paper Shao et al. (2021), the regression network comprises two layers. Each layer begins with a convolutional unit, followed by batch normalization and ReLU activation. A final fully-connected (FC) layer performs the regression of parameters for the geometric transform, outputting the affine matrix $\theta$. The matrix $\theta$ serves as the affine transformation for the registration process.

## 2.4 Loss Function for Feature Matching

The loss function was determined as the sum of squared differences (SSD) between the original input MRI and the deformed pathology image. The formula of the loss function was shown as:

$$loss = \sum_{i}^{H} \sum_{i}^{W} \|I_A(i, J) - I_B(i, J) \bullet \phi_\Theta(i, j)\|^2 \tag{2}$$

where $\phi_\Theta(i, j)$ is the related transformation vector from the output of feature matching sub-network, $H$ is the height of the image and $W$ is the weight of the image.

## 2.5 Fusing Pathology and MRI

After completing the image registration process, the pathology images, MRI scans, and annotated lesions were aligned with the corresponding MRI slices using the estimated composite affine transformation $\theta$. It is important to note that pathology images typically have larger dimensions than sliced MRI images, which are usually $512 \times 512$. Consequently, the

deformed pathology images maintain their original size as high-resolution images, as the affine transformation is only applied to the original image. Once the affine transformation matrix $\theta$ is determined based on the input image, it remains fixed. During inference for high-resolution pathology images in gigapixel scale, we first resize them to the same smaller size as the MRI slides. These resized images are then fed into our network to obtain the transformation matrix $\theta$. Finally, we apply $\theta$ to the original resolution pathology image to obtain the registered high-resolution pathology image.

## 3 Evaluation

In this section, we demonstrate the effectiveness of our proposed registration framework on two prostate cancer datasets Madabhushi A (2018). Our evaluation consists of three parts. In the first part, we evaluate the fusion image from the resolution comparison, and Structural Similarity (SSIM). In the second part, we will display some qualitative results for our fusion image. In the last part, we will have a quantitative evaluation of downstream lesion segmentation tasks.

### 3.1 Comparison of the Fused Images and The Original MRI Images

We evaluated the difference between the fusion image and the original MRI both for resolution enhancement, and SSIM on the prostate cancer dataset Madabhushi A (2018).

#### 3.1.1 Resolution Enhancement

The resolution enhancement from the original MRI to the fused images for each test case is from 320*320 MRI to a resolution of 2000*2000, which is  39x times higher resolution.

#### 3.1.2 Structural Similarity Index (SSIM)

We also evaluated the structural similarity of the fusion image and MRI image by Structural Similarity Index (SSIM). SSIM is a metric used to measure the similarity between two images. The calculation of SSIM is located at Appendix B. The SSIM value ranges from -1 to 1, where a value of 1 indicates perfect similarity between the two images. The result of SSIM in all six cases is shown in Fig. 2. As the value shows, the lowest SSIM score we have for aaa0069 is still 0.933 which is close to the max SSIM value which is 1. The average SSIM is 0.939. That suggested that our fusion image has high similarity with the up-scaled MRI in both luminance, contrast, and structure.

### 3.2 Quantitative and Qualitative Result on Downstream Segmentation Task

#### 3.2.1 Quantitative results

We assessed the quantitative results for lesion segmentation using two metrics: the Euclidean distance and the Dice score, applied to MRI and registered pathology images across six test cases. A higher Dice score indicates increased similarity, while a lower Euclidean distance(ED) signifies reduced separation. In this evaluation, lesions manually annotated on both MRI and pathology images served as landmarks for assessing the segmentation quality. These lesion annotations were solely used for evaluation purposes and were not

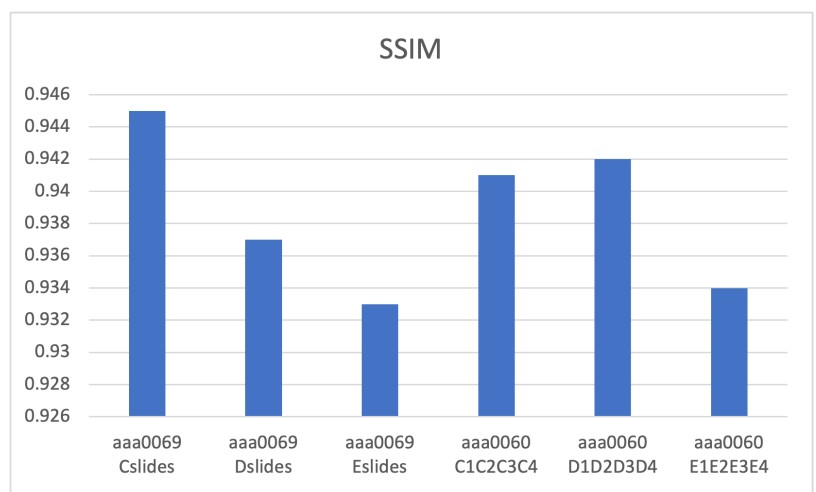

Figure 2: SSIM score of all six test cases

incorporated into the training process. To ensure consistency, the same transformation was applied to the original lesions on the pathology images. The Euclidean distance measured the distance between landmarks on MRI and registered pathology images, while the Dice score indicated the overlap of lesions between the two images. The initial findings, summarized in Table 1, revealed an average Dice score of $64.0\% \pm 4.1\%$ across the six cases, with an average Euclidean distance of $2.074, \mathrm{mm} \pm 0.776, \mathrm{mm}$. Compared to the original work by Shao et al. Shao et al. (2021), our method demonstrated improvements, increasing the average Dice score from $57.3\%$ to $64.0\%$ and reducing the average distance from 5.42 mm to 2.074 mm. Since the original paper did not report the Dice score and Euclidean distance (ED) for each case, our comparison in this paper focuses on the mean Dice score and mean ED across all cases.

| Case ID | Euclidean Distance (mm)↓ | | Dice ↑ | |
|---|---|---|---|---|
| | Original | Our | Original | Our |
| aaa0060 C1C2C3C4 | N/A | 1.663 | N/A | 0.653 |
| aaa0060 D1D2D3D4 | N/A | 1.872 | N/A | 0.632 |
| aaa0060 E1E2E3E4 | N/A | 1.763 | N/A | 0.665 |
| aaa0069 CSlides | N/A | 1.532 | N/A | 0.681 |
| aaa0069 DSlides | N/A | 2.851 | N/A | 0.600 |
| aaa0069 ESlides | N/A | 2.767 | N/A | 0.609 |
| Mean | 5.42 | **2.074** | 0.573 | **0.640** |

Table 1: Quantitative result of registering pathology images with MRI image on all the six test cases. The Euclidean Distance and the Dice score metrics are calculated between the ground truth of lesions on MRI and fused pathology images.

### 3.2.2 QUALITATIVE RESULTS

We present the visualization of our registered results in Fig. 3. The figure showcases the following components: the upscaled MRI in section (a), the pathology image with the related mask in section (b), the fusion image in section (c), and the lesion labels in section (d).

Each row corresponds to a visualized patient, with cases aa0069 Cslides, aa0069 Dslides, and aa0069 Eslides displayed from top to bottom, respectively. The lesion labels were annotated by an expert pathologist reported on the original dataset. We applied the same transformation to the labels and registered them with the fusion image, as illustrated in sub-figures (d) and (e).

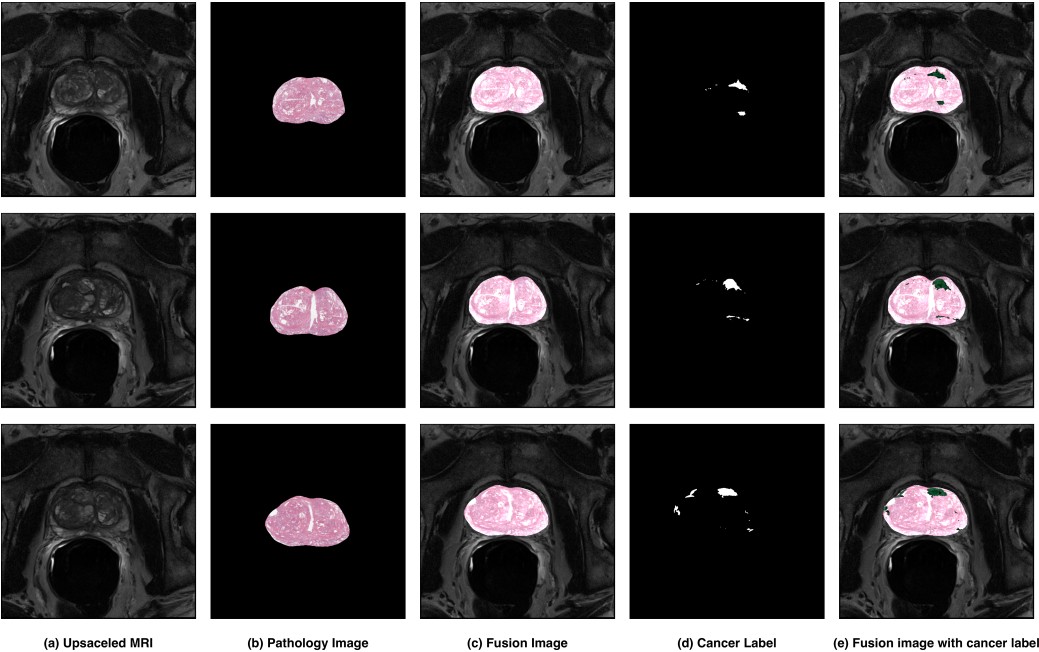

(a) Upsaceled MRI    (b) Pathology Image    (c) Fusion Image    (d) Cancer Label    (e) Fusion image with cancer label

Figure 3: Visualization of Qualitative Result. (a) is the visualization of three MRI sample slides that aligned with the axial view and resized to (2000,2000). (b) is the visualization of pathology images. All sides were combined by four WSIs. (c) is fused pathology-MRI image. (d) is the visualization of registered lesions segmentation. (e) is the visualization of registered lesions with fusion images.

## 4 Conclusion, Limitation and Future Work

In this paper, we proposed a registration framework for aligning radiological and pathological images. Through our experimental analysis, we have demonstrated the capability of self-supervised learning approach to bridge the resolution gap between these two modalities, facilitating accurate registration without the need for labeled training data. The fusion of high-resolution pathology data with low-resolution radiological scans have shown great promise in enhancing the diagnostic potential of medical imaging, particularly in tasks such as cancer segmentation. However, it is important to acknowledge the limitations of our work. Our experiments were conducted on limited datasets, and the fused high-resolution image was not directly generated but created through the registration process. These limitations highlight areas for future research and improvement. Moving forward, further research and validation studies will be essential to validate the clinical utility and robustness of our framework across diverse datasets and clinical scenarios.

## Appendix A. Mutual Information(MI)

We also evaluated the similarity of fusion image and MRI image by Mutual Information (MI). MI is a measure of the mutual dependence between two random variables. It quantifies the amount of information obtained about one random variable through the other random variable. In the context of image processing, MI can be utilized to assess the similarity between two images by comparing the statistical dependencies between their pixel intensities. The mutual information between two discrete random variables $X$ and $Y$ is defined as:

$$I(X;Y) = \sum_{x \in X} \sum_{y \in Y} p(x,y) \log \left( \frac{p(x,y)}{p(x)p(y)} \right)$$

where: $I(X;Y)$ is the mutual information between $X$ and $Y$. $p(x,y)$ is the joint probability mass function of $X$ and $Y$. $p(x)$ and $p(y)$ are the marginal probability mass functions of $X$ and $Y$, respectively.

Higher MI values indicate a greater similarity between the images, while lower MI values suggest dissimilarity. The result of MI information is show in the table 2. As shown in the table, our MI score is close to the upper bounder of MI score which is 4 on average. We also display the normalized MI score as a reference. The score suggested our fusion image has high similarity to the upscaled MRI.

| Case ID | Mutual Information ↑ | | |
| --- | --- | --- | --- |
| | MI | MI Upper boundary | MI(Normal) |
| aaa0060 C1C2C3C4 | 4.112 | 15.6 | 6.80e-7 |
| aaa0060 D1D2D3D4 | 4.142 | 15.6 | 6.82e-7 |
| aaa0060 E1E2E3E4 | 4.033 | 15.6 | 6.71e-7 |
| aaa0069 CSlides | 4.161 | 15.6 | 6.93e-7 |
| aaa0069 DSlides | 4.011 | 15.6 | 6.69e-7 |
| aaa0069 ESlides | 4.032 | 15.6 | 6.70e-7 |

Table 2: MI information and score of all six test cases.

## Appendix B. SSIM

SSIM takes into account three aspects of image quality: luminance, contrast, and structure. The formula for SSIM is given by:

$$\text{SSIM}(x,y) = \frac{(2\mu_x\mu_y + c_1)(2\sigma_{xy} + c_2)}{(\mu_x^2 + \mu_y^2 + c_1)(\sigma_x^2 + \sigma_y^2 + c_2)} \tag{3}$$

where: $\mu_x$ is mean of x, $\mu_y$ is mean of y, $\sigma_x$ is standard deviation of x, $\sigma_y$ is standard deviation of y, $\sigma_{xy}$ is covariance of x and y, $c\_1$ is constant to stabilize the division with weak denominator, $c\_2$ is constant to stabilize the division with weak denominator.

## Appendix C. Ablation Study

In our methodology, we utilized the DINO self-supervised learning (DINO-SSL) framework for pre-training on diverse datasets. To measure the effectiveness of our pre-training strategy using self-supervised learning, we employed the downstream lesion segmentation task as our evaluation metric. The results are summarized in Table 3. All experiments utilized the same architecture, leveraging the vit small backbone within the DINO framework with an output dimension of 348. The first row of the table corresponds to our framework without pre-training, trained with feature matching network on two training datasets. Here, we observed a Dice score as low as 55%. However, when employing pre-trained model weights from ImageNet, as provided by the official DINO paper, for both MRI and pathology feature extractors, we observed a notable improvement, with the Dice score increasing to 61%. Furthermore, adopting separate pre-training on TCIA and TCGA datasets for MRI and pathology images can led to further enhancement, with the Dice score reaching 64%. This highlights the significance of pre-training on distinct datasets as a crucial factor in enhancing the performance of existing methodologies. Through our ablation study, we identified key factors contributing to the overall efficacy of our proposed framework.

| Arch | SSL Method | Dataset | Epochs | Dim | Dice Score |
|------|-----------|---------|--------|-----|-----------|
| ViT-S/16 | Dino | scratch | N/A | 348 | 0.55 |
| ViT-S/16 | Dino | ImageNet | 100 | 348 | 0.61 |
| ViT-S/16 | Dino | TCIA/TCGA | 100 | 348 | 0.64 |

Table 3: Dice Scores of Various Training Configurations using Pre-Trained Feature Extractors on Different Datasets for Downstream Prostate Cancer Segmentation

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
