# OpenReview forum: "Upscaling Prostate Cancer MRI Images to Cell-level Resolution Using Self-supervised Learning"
_MICCAI.org/2024/Workshop/COMPAYL — COMPAYL 2024_

### Official Review · Reviewer_GELK · 2024-07-09
**Review of paper 24: A method for MRI-HE registration**

**Custom Rating:** 2
**Confidence:** 5

**Review:**

The authors proposed a method for the registration of MRI and HE images. However, the paper is not well-written, and many sections are unclear. The proposed framework lacks significant novelty, as both modules of the network are adopted from existing works. Additionally, the results are reported using a limited dataset and inadequate evaluation metrics.

Pros:
- The paper is well-structured

Major Comments:
- There is a wide literature on the registration of MRI and histology images and the use of deep learning for image registration. The authors need to rewrite the introduction and cite relevant literature.

- In the contributions section, point three is a claim that self-supervision helps with an insufficient dataset, which is not entirely accurate, and they lack experiments to support this claim.

- In equation 2, what is J? The authors also need to clarify whether the dot in the formula represents an inner product or something else.

- The size of the test dataset is very small (6 cases) and cannot adequately reflect the efficiency of the proposed framework. The authors need to test the model on a larger dataset, as evaluating segmentation on only 6 cases is not significant.

Minor comments:
- Table 1 can be restructured as two columns containing N/A values.
- Section 2.2 can be made easier to understand by providing more explanation about the work from which this module is adopted (DINO).

---

### Official Review · Reviewer_FWpe · 2024-07-12
**A hard problem, an interesting solution, but what application?**

**Custom Rating:** 3
**Confidence:** 5

**Review:**

The authors propose a deep learning-based approach to affinely register high-resolution histopathological images with MRI images of the prostate. Their method employs two domain-specific feature extractors pretrained in a self-supervised manner, alongside two correlation mapping and feature matching modules to predict the registration parameters. Despite the significant challenge of addressing a large domain shift, their method has shown promising results, highlighting the potential of self-supervised learning for feature representation relevant to registration. However, there are several areas for improvement and some questions that need to be addressed.

M1 - The first question is about the application of this task. The authors suggest "enhancing cancer evaluation," but the specifics are unclear. Are they referring to tasks like tumor segmentation in MRI images? If the prostate tumor has already been resected for histology, what is the benefit of segmenting the tumor in an MRI image? In this scenario, the ground truth is the existing whole-slide image (WSI), so there is no need to revisit the MRI image. If the goal is to create a reliable dataset for tumor segmentation in MRI scans, this should be discussed in the manuscript. Additionally, it should be demonstrated that this method is superior to manual annotation by radiologists on MRI images. Authors should clearly mention or cite relevant publications about the application of this task.

M2 - Similar to the previous comment, the need for "fused" images is unclear. The fused images appear to be histology images transformed to fit MRI images. In clinical use, why not directly examine the histology of WSIs?

M3 - The manuscript lacks clarity in certain methodological explanations. For example:
- In section 2.5, it states that MRI scans are aligned with corresponding MRI slides, which seems to be a typo.
- In section 2.5, the phrase "During inference for ..." is poorly written. It would be better to mention the resizing step in the feature extraction or matching sections and then explain how the obtained transformation function is applied to the original high-resolution histology image. Also, clarify if the WSI was transferred in a patch-wise manner.
- In section 2.1, it mentions that MRI slides and pathology images were aligned. Was this done manually?
- In section 2.4, it is unclear how the ground truth transformation vector is obtained for use in the loss function.

M4 - Given the small test dataset, why not use cross-validation (each time using different parts of the MRI-Prostate dataset for testing) to gain a more robust understanding of the model's performance?

M5 - Is it possible to use existing "Foundation models" for feature extraction instead of pretraining the models? This should be discussed or mentioned in the limitations/future work section.

Furthermore, there are some minor comments:

m6 - It is common to use present tense in scientific writing. For example, on page 3, the phrase "Our method advanced the state-of-the ..." should use "advances" to align with the rest of the manuscript.

m7 - Citations are used inappropriately throughout the manuscript. When a citation is the object or subject of a sentence, it should be used outside of parentheses, e.g., "Shao et al. (2021) found that ...". When a citation is used to refer to a paper, it should be in parentheses, e.g., "In the MRI-Prostate dataset (Madabhushi et al., 2016)...". The authors should revise the manuscript to correct citation formatting.

m8 - In Table 1, instead of naming the baseline "original," cite the publication or name of the method, or simply call it "baseline."

---

### Official Review · Reviewer_7c7A · 2024-07-15
**Review for "Upscaling Prostate Cancer MRI Images to Cell-level Resolution Using Self-supervised Learning"**

**Custom Rating:** 3
**Confidence:** 3

**Review:**

The authors propose a novel self-supervised deep learning framework that integrates pathology WSIs with MRI images, creating hybrid images with detailed cancer tissue structures. The multi-modal pathology/MRI learning is an underresearched area and there are some interesting ideas in this paper.

However, there are multiple shortcomings to the study overall:

1)There is no comparison to any baseline or state-of-the-art.

2)Would be useful to also report the Euclidean distance between landmarks of anatomical structures i.e urethra and not only lesions.

3)The correlation mapping process used to align MRI and pathology images might result in ambiguous matches, especially when dealing with highly heterogeneous tissue structures. This ambiguity could affect the accuracy of the registration and the quality of the fused images.

4) As the test set size is very small it would be useful to see on which areas does the segmentation improves with the help of the proposed method.

5) The authors do not mention anything about making the code public.

6) The authors report SSIM, Euclidean distance and Dice for each case and even make a bar plot for SSIM of each of the 6 testing cases. It is usually enough to report an average metric with a standard deviation / median per dataset.

---

### Decision · Program_Chairs · 2024-07-16

Accept